Model for finding the number of honey bee colonies needed for the optimal foraging process in a specific geographical location

Komasilova Olvija
Komasilovs Vitalijs
Kviesis Armands
http://orcid.org/0000-0002-6974-8653 Zacepins Aleksejs aleksejs.zacepins@llu.lv
Department of Computer Systems, Faculty of Information Technologies, Latvia University of Life Sciences and Technologies , Jelgava , Latvia
Colla Sheila
Electronic publication date: 2021 Sep 17
Publication date: 2021
Volume: 9
Electronic Location ID: e12178
Received 2021 May 24; Accepted 2021 Aug 29
Copyright: © 2021 Komasilova et al.
Copyright year: 2021
Copyright holder: Komasilova et al.
License: This is an open access article distributed under the terms of the Creative Commons Attribution License, which permits unrestricted use, distribution, reproduction and adaptation in any medium and for any purpose provided that it is properly attributed. For attribution, the original author(s), title, publication source (PeerJ) and either DOI or URL of the article must be cited.
License URL: https://creativecommons.org/licenses/by/4.0/

Keywords: Smart apiary, Honey bee, Precision beekeeping, Precision apiculture, Bee colony foraging process, Hiveopolis

Funding: European Union’s Horizon 2020 research and innovation programmes 824069 Hiveopolis project has received funding from the European Union’s Horizon 2020 research and innovation programmes under grant agreement No. 824069. The funders had no role in study design, data collection and analysis, decision to publish, or preparation of the manuscript.

==============================
Finding a proper location for a bee apiary is a crucial task for beekeepers and especially for travelling beekeepers. Normally beekeepers choose an appropriate apiary location based on their previous experience and sometimes the location may not be optimal for the bee colonies. This can be explained by different flowering periods, variation of resources at the known fields, as well as other factors. In addition it is very challenging to evaluate how many bee colonies should be placed in one geographical location for an optimal nectar foraging process. This research presents a model for finding the number of honey bee colonies needed for the optimal foraging process in the specific location, taking into account several assumptions. Authors propose to take into account potential field productivity, possible chemical contamination, surroundings of the apiary. To run the model, several steps have to be completed, starting from the selection of area of interest, conversion to polygons for further calculations, defining the roads in the selected area. The outcome of the model number of colonies that should be placed is presented to the user. The Python language was used for the model development. The model can be extended to use additional factors and values to increase the precision of the evaluation. In addition, input from users (farmers, agricultural specialists, etc.) about external factors that can affect the number of bee colonies in the apiary can be taken into account. This work is conducted within the Horizon 2020 FET project HIVEOPOLIS (Nr.824069).

Introduction

The Western honey bee, Apis mellifera plays a crucial role as a pollinator worldwide (Bolshakova & Niño, 2018). Although during the last few years honey bee populations are steady and increasing in numbers with some fluctuation, there are a number of threats responsible for honey bee health and survival (Neov et al., 2021). The decline of pollinators may have important ecological and economic impacts that could significantly affect the maintenance of wild plant diversity, crop production and human welfare (Lázaro et al., 2016).

With the help of information and communication technologies by applying constant remote monitoring of the bee colonies, it is possible to react on-time to unpredictable events within the colonies (Komasilovs et al., 2019). Precision beekeeping (sometimes called also precision apiculture) is emerging and is defined as an apiary management strategy based on the remote and real-time monitoring of individual bee colonies to minimize resource consumption and maximize the productivity of bees (Zacepins et al., 2015). One of the important management decisions for the beekeeper is to choose how many hives should be placed in one location for optimal foraging performance. This question is also important for the commercial beekeepers as they generally own many colonies of honey bees, and if the number of colonies at one place is excessive, the foraging competition among colonies will have a direct and negative effect on hive yield (Akratanakul, 1990). In this study the optimal foraging performance is understood as follows: optimal number of colonies placed at one geographical location (within one apiary) should be able to forage the maximal amount of potential resources available during the whole active foraging period, or when considering migrating beekeeping, during one specific blooming time.

Evaluation of the number of hives needed to be placed in a specific location usually is based on beekeepers’ experience or the limited availability of physical space (Komasilova et al., 2020). Beekeepers can also be informed by the local farmers about the pollination service needed and then beekeepers can decide how many hives should be transported there. In some countries, for instance in Indonesia (Gratzer et al., 2019), migratory beekeeping is very common, and beekeepers are forced to change the apiary location very often to provide food sources for their colonies. To have more precise evaluation on the number of needed colonies there are some researches in agricultural pollination which provides guidelines for stocking honey bee colonies in a crop to maximize pollination in the most economical way (Delaplane et al., 2013). The stocking rates of colonies required can be estimated indirectly by using the pollination potential of individual foragers and extrapolating the number of foraging honey bees and colonies required to pollinate a crop (Goodwin et al., 2011). In fact, most pollination handbooks are based on recommendations of a particular number of honeybee hives per crop-cultivated area (Free, 1993). Thus, evaluation of the optimal number of bee colonies is an important question, as aspects such as in-between colony resource distribution and competition should be also considered. Competition may occur among nearby bee colonies. The area of competition is usually in a circular band with radius close to the average flight distance. As the hives become further apart, competition decreases (Esteves, Villadelrey & Rabajante, 2010). The occurrence of imbalances between the number of bee colonies in one location and the important bee forage areas and the subsequent decline in productivity per colony has been observed (Khanbash, 2001). For example, natural colony density as determined for both Palearctic and Nearctic forests was established at 0.5 colonies per km2 (Galton, 1971; Visscher & Seeley, 1982). Other research shows that the number of natural density is 0.11–0.14 honey bee colonies per km2 (Oleksa, Gawroński & Tofilski, 2013; Kohl & Rutschmann, 2018). For sure natural honey bee colonies densities that occur naturally in forests can vary widely from biome to biome and from agroecosystem to agroecosystem. In contrast to natural colonies many recommendations for orchard pollination call for one or more two-story colonies per acre (http://treefruit.wsu.edu/orchard-management/pollination/honey-bees/). In most cases, success in beekeeping depends on the availability of sufficient bee foraging resources in terms of both quality and quantity of nectar and pollen (Addi & Bareke, 2019). Hence, beekeeping is more dependent on the existing natural resource conditions of an area than any other livestock activities (Al-Ghamdi et al., 2016).

The aim of this research is to present a developed model, which can provide support for the beekeepers in finding an optimal number of hives which could be placed in one geographical location for the effective foraging process.

This model is based on the previous author model for the bee apiary location evaluation which is described in (Komasilova et al., 2020) but has another point of view and more sophisticated level of details.

This research is conducted within the Horizon 2020 FET programme project HIVEOPOLIS (https://www.hiveopolis.eu/). Collection of hives, technologies and humans is called Hiveopolis in our concept. HIVEOPOLIS technology will be integrated in a way that it provides a synergistic added value to the colony, to its owner and to the society in general.

Materials and Methods

This section describes a proposed model development process for finding the optimal number of hives in the specific geographical location which is chosen by the beekeeper and the used parameters. The model’s development process is divided into two main steps. In the first step, the aerial image of the region is annotated with polygons for foraging fields and lines for roads, and an estimated value of resources on those fields are made. As the result, authors obtain a semantically annotated map, which can be used for further automatic evaluation of a specific apiary location. Based on this semantic map, in the second step, chosen location is evaluated using additional levels for pesticides, road effect on the foraging activity. Then the exact number of hives which can be placed in the location is presented. As well several predefined parameters are used for the calculations.

For the final calculation of an optimal number of colonies that should be placed at a specific location chosen by the beekeeper several additional parameters should be introduced.

(1) delta=h∗k=2/5∗0.35

where:

k-honey bees foraging efficiency (k = 0.35 used in example);

h-nectar to honey production rate (h = 0.4 used in example);

This means that bee colony needs five kg of nectar to produce two kg of honey (Grebennikov, 2005), and honey bee foraging efficiency is 35% (Narchuk & Moreva, 2016). Foraging efficiency describes the amount of possible percent of nectar foraging as all nectar could not be foraged. Honey bees are not able to forage all nectar due to the several reasons: part of the nectar will be collected by other insects; bees will not be able to visit all flowers; due to climatic conditions that potentially could be unfavorable.

Hcolony = 90

For one bee colony approximate 90 kg of honey is needed for self-consumption (Lebedev & Krivtsov, 2019).

Haverage = 60

Defined amount of honey beekeeper would like to collect from one bee colony. This parameter can be individually defined by the beekeeper.

(2) H(x,y)≥Hcolony

where:

H(x, y)-total potential foraging resources for single colony placed at (x, y);

Hcolony = amount of honey needed for the colony survival

C1 = field nectar productivity coefficient = {0.2 0.4 0.6 0.8 1.0}

C2 = field pesticide contamination coefficient = {−1.0 −0.5 0}

Cf–field overall coefficient taking into account the nectar productivity and pesticide contamination

(3) Cf={ifC1+C2>0thenC1+C2else0}

C3 = coefficient for field availability = {1 0.7 0.5}, depending on the road type

C4 = coefficient for the pollution level near roads {−0.5 −0.3 −0.1}

Spot1–area of the field segment falling into the area of nectar collection, without taking into account road coefficient

S2–area of the field segment falling into the area of nectar collection and located near the road (taking into account road coefficient)

H(i)-amount of nectar foraged from field segment i into the area of nectar collection

Hlosses(j)-the amount of honey losses from field j, taking into account areas near roads

(4) H(i)=Cf∗C3∗Spot∗delta

(5) Hlosses(j)=Cf∗C3∗(−C4)∗S2j∗delta

Total nectar collected from the are can be calculated by the formula below:

(6) H=∑i⁡H(i)−∑J⁡Hlosses(j)

To calculate the number of total colonies that should be placed in the apiary formula below can be used:

(7) N(x,y)=H(Hcolony+Haverage)

where:

N(x, y)-number of colonies placed at a location (x, y)

Proposed model is developed in Python language using several libraries, including Matplotlib for creating static, animated and interactive visualization in Python (Hunter, 2007), NumPy (van der Walt, Colbert & Varoquaux, 2011) and Shapely package for manipulation and analysis of planar geometric objects (Gillies, 2007).

Below the proposed model is described and illustrated in detailed step-by-step process.

Getting the map of the target location

At the first stage it is needed to choose the target location for the apiary and crop the region of interest to work with (see Figs. 1A and 1B). Authors used Google Maps for the image selection. Used part of the map can be seen here:

Figure 1 Selected region for the model development.

Chosen geographical region, which is used for model development. (A) Selected region for the model development. (B) Annotated region for the model development.

https://www.google.com/maps/@56.4611975,22.9313681,6580m/data=!3m1!1e3

It is assumed that the dimension of the region of interest is 10 to 10 km. Region of interest can be also different if needed.

W = 10,000

where: W–map region width (m)

Definition of foraging places and roads

As foraging places agricultural fields are considered at this point. In the future also individual gardens, parks and other locations can be used. Agricultural fields are represented by different polygons and it is needed to mark all of them. At this moment, this task is completed manually using the author’s developed web interface.

User should mark all the vertices of each polygon (field) and the tool will extract their coordinates.

In the example, there are 201 polygons defined within the selected region of interest.

(8) ∀fn∈F→Pn=O(fn)

where:

F–fields from regions of interest

Pn–number of polygons

O(f)–polygon outlining the field

In addition, lines which are representing roads (or impossible sources of resources for bees) are defined. The lines are marked manually using the same web interface. User should mark all the end (key) points of each line and the tool will extract their coordinates. In the example, there are 28 lines defined within the selected region.

(9) ∀rn∈R→ln=O(rn)

where:

R–roads from regions of interest

ln–number of lines

O(r)–line representing the road

Transforming the real image to semantically annotated map of polygons and lines

Based on the coordinates generated in the previous stage, the region annotated with chosen polygons and roads is generated (see Fig. 2). This region afterwards can be processed in different ways, applying different layers and parameters for the polygons.

Figure 2 Generated digitized map of the region of interest with fields.

Digitized map of the chosen geographical region. All annotated polygons are digitized. (A) Generated digitized map of the region of interest with fields. (B) Generated digitized map of the region with roads.

After the digitized map is built several layers (levels) are introduced to the proposed model.

Level 1-definition of the nectar production

Resource availability at the agricultural fields is one of the main parameters for the foraging resource evaluation in the specific geographical location. In the real situation available foraging resources should be related to the theoretical amount of resources available for bee colony forage. It is a very challenging task to evaluate the exact amount of resources available at a foraging location. It is possible to use the information about specific plants and crops and their indices describing pollen and/or nectar production. In the provided example, polygons’ values are assigned randomly to demonstrate the calculation method itself, therefore values can be different from the real situation. It is assumed that surrounded fields can have five different foraging coefficients, and those coefficients are related to the nectar production from one hectare. We assume that nectar production is from 20 to 100 kg and coefficients are from 0.2 to 1.0:

(10) vn=V(pn)=U∈{0.2,0.4,0.6,0.8,1.0}

where:

V(p)-plant nectar production index in a given field (represented by polygon)

U-uniform random distribution over a set of values

For better visualization fields are color encoded. Colors are ranging from bright red to bright green. The region with the highest value will have its polygon colored bright green (see Fig. 3).

Figure 3 Encoding of the fields based on a potential nectar production.

Fields are color encoded. Colors are ranging from bright red to bright green. The region with the highest value will have its polygon colored bright green.

Level 2-definition of the possible pesticides contamination

One of the main ecological anthropogenic factors which has a negative effect on bee colony vitality is chemical (pesticides, herbicides) treatment of the plants. As a result of those operations bees are foraging and taking to the hive nectar and pollen contaminated with pesticides, which can cause poisoning, decrease the honey bee natural immunity and at the end lead to the death of the whole colony (Frazier et al., 2015). Beekeeping products are also contaminated with toxic substances and can negatively affect human health.

Therefore additional parameter was implemented in the model as a coefficient that represents the potential contamination of the fields.

It is possible to use the information about the frequency of treatment of a particular field with chemicals. This information can be collected from the farmers, or farmers can provide this information using some specific app or web platform. In the provided example, polygons’ (fields) values are assigned randomly to demonstrate the calculation method itself. It is assumed that three different field states can be in the selected region. The field can be with a high risk of contamination (coefficient value is equal to −1.0). In that case field productivity is decreased to 0.0. If contamination coefficient is equal to 0.5, then productivity rate is decreased to 0.5. In an ideal situation, when the field is not processed with chemicals, the contamination coefficient is equal to 0.0. Contamination coefficients are defined as follows:

(11) dn=D(pn)=U∈{−1.0,−0.5,0.0}

where:

D(p)-chemical contamination index in a given field (represented by a polygon)

U-uniform random distribution over a set of values

For better visualisation fields are color encoded depending on their pesticides coefficient value, ranging from bright red to bright green, with three colour steps. The region with the zero value (field without pesticides) will have its polygon coloured bright green (see Fig. 4).

Figure 4 Encoding of the potential volume of pesticides in the fields.

Fields are color encoded depending on their pesticides coefficient value, ranging from bright red to bright green, with three colour steps. The region with the zero value (field without pesticides) will have its polygon coloured bright green.

Level 3-definition of each road value

When choosing the apiary place it is needed to consider that proximity of roads has a negative effect on bee colony foraging behaviour. This can be explained by the fact that foraging resources are polluted by the transport emissions and heavy metals. As well bees can die due to the collision with the transport if they fly across the road. Some authors suggest to place the apiaries at least 500 m from the highways and railways (Solovieva, 2009), if the colony will be used for pollen production, then distance from highways should be more than one km (Аnoshkina, 2018). Other scientists claim that 20–30 m is enough (Mannapov et al., 2015). For instance in the Civil Law of Latvia (article 1101, https://likumi.lv/ta/en/en/id/225418-the-civil-law) it is stated that bee colonies may be placed, in rural areas at least 15 m, but in cities, towns or villages, at least 25 m from traffic routes or the land boundaries of neighbors, such distance being calculated from the center of the hive to the edge of the road or the boundary, and if the apiary is fenced–in rural areas with at least a two m, but in cities, towns or villages, a two and a half m high close-set fence or hedge–the bee colonies may be placed without regard to the aforementioned distances.

We propose to classify roads into three groups and assign coefficients based on required distance from the road. Let’s assume that minimal distance from the road to apiary is dependent on the road coefficient. Highway with intensive traffic has a coefficient 0.5 (distance to the apiary should be more than 500 m). Railroads has the coefficient 0.3 (distance to the apiary should be more than 300 m) and the country roads with light traffic have the coefficient 0.1 (distance to the apiary should be more than 100 m).

In the provided example, roads’ coefficient values are assigned based on the real situation. It is assumed that three different road types are in the selected region.

(12) rn=V(ln)=R∈{−0.5,−0.3,−0,1}

where:

V(l)-road index (represented by polygon with width)

R-roads from regions of interest

To visually differentiate roads by their value, color and line width encoding is implemented, ranging from bright red to bright green, with three color steps. The region with the highest value (less harmful, as coefficients with minus sign) will have its line colored bright green (see Fig. 5).

Figure 5 Encoding of the roads on a selected area.

Roads are encoded by their value, color and line width encoding is implemented, ranging from bright red to bright green, with three color steps. The region with the highest value (less harmful, as coefficients with minus sign) will have its line colored bright green.

In addition, it is considered that crossing the road can increase the probability of bee death affecting bee colony productivity due to decrease of the number of bee foragers. It is needed to define the active polygon, where apiary is located and non-active, which are located across any road (railway, railroad or country road) (see Fig. 6).

Figure 6 Digital map with the active polygon between all roads map (for point with coordinates (552, 364)).

Active polygon indicates the location of bee colonies.

For those polygons potential field productivity is decreased to 70%. Depending on the road type field productivity will be additionally decreased by 20% for the highway and railroad. Digital map with the polygons between main roads map is demonstrated in Fig. 7.

Figure 7 Digital map with the polygons between main roads.

Level 4-bee colony flight area

To represent bee colony foraging area, several parameters should be pre-defined, such as target location, expressed as a point coordinate (x, y), and flight area radius (in km). The parameters can be user adjusted. This information is defined as a separate layer for the model. As an example, Fig. 8 represents a bee colony foraging area at location (552, 364) with bee flight radius three km (Prešern, Mihelič & Kobal, 2019).

Figure 8 Target location of the apiary with the potential bee colony flight radius.

Image represents a bee colony foraging area at a specific location (coordinates: 552, 364) with a bee flight radius of three km.

Layer intersections with the colony flight area

For the model calculations total areas of regions of intersections with the colony flight area were used. Different layer parameters were taken into account to determine potential production value. Thus for the given example of the apiary location in the point (552, 364) only polygons shown in Figs. 9A–9E below are considered and evaluated. If the foraging field is situated within active polygon and it is not needed to cross any roads, then field coefficient is equal to one. If a big road should be crossed then coefficient becomes 0.5.

Figure 9 (A) Intersection of bee colony flight area with the field map with the field coefficients. (B) Intersection of bee colony flight area with the field map taking into account pesticide coefficients. (C) Intersection of bee colony flight area with the road layer. (D) Intersection of bee colony flight area with the polygon map between all roads displaying an active polygon. (E) Intersection of bee colony flight area with the polygon map between only main roads displaying an active polygon.

Calculation of a number of colonies to be placed at a specific location (x,y)

For the given example the total amount of honey that can be produced in the given location is 7688.84 which means that 51 honey bee colony could be placed in that location. The result can also be written in JSON format and used by other services for further application or comparison between the locations.

{

"x": 552,

"y": 364,

"onRoad": false,

"honey": 7688.84,

"hives": 51

}

Results and Discussion

The authors presented approach and model for finding the number of honey bee colonies needed for the optimal foraging process in the specific geographical location. Despite the fact that this is an actual and important question for beekeepers there are not many scientific references dealing with this problem and providing different models for such evaluation taking into account various factors and variables. As mentioned in the introduction usually beekeepers decide on the amount of the hives based on previous experience without sophisticated analysis of other factors, like availability of foraging resources, presence of nearby roads and other limitations.

There are researches in agricultural pollination which provides guidelines for stocking honey bee colonies in a crop to maximize pollination in the most economical way (Delaplane et al., 2013), but not every beekeeper is using those guidelines.

A less common approach to evaluate the number of colonies is to stock fields with different numbers of colonies and establish whether the rates used have an effect on pollination (Palmer-Jones & Clinch, 1974; Vaissière, 1990; Brault, De Oliveira & Marceau, 1995). But this approach is very rare because it is difficult to obtain acceptably large numbers of replicate fields for each treatment.

Some researches show that crop pollination levels appear to be optimal in real-world systems (Pfister et al., 2017) the greater weight of evidence suggests that current pollination levels are usually suboptimal (Garibaldi et al., 2020). This acknowledges the importance and need of some detailed models for the optimal bee colony number evaluation.

For all practical purposes, an apiary should normally consist of 30 to 80 colonies, the two most important factors determining the optimum number being the worker population in the hives and the amount of forage available in the area. Where nectar and pollen sources are abundant, an apiary can consist of 50 or more hives with relatively large worker populations without endangering satisfactory yields (Akratanakul, 1990).

The authors present the model for the evaluation of the foraging location and combining it with real crop nectar production values in the future will provide a valuable tool for practical beekeepers.

Conclusion

In this study we propose a computational model that can be used by beekeepers to find the optimal number of hives that should be placed in one apiary location for effective resource foraging process.

To determine the number of hives in a specific location, the developed model incorporates multiple factors that have a positive and negative effect on foraging-such as the number of fields in the area of interest, field productivity, possible level of contamination (e.g., pesticides), specifics of surroundings, like main roads, railways.

The proposed model is implemented in Python language and potentially can be improved in the future by adding additional levels to better describe the real-life situation/environment.

For future work, we plan to automate the map-to-polygon transformation, to facilitate the user in data pre-processing for the model.

The proposed model requires further evaluation and validation in real world conditions. However, this remains as a challenging task, involving multiple hives, locations and the evaluation of productivity per region.

Supplemental Information

Supplemental Information 1 Developed model in Python.

Python scripts developed to describe and run the model

Click here for additional data file.

Additional Information and Declarations

Competing Interests

Author Contributions

Data Availability

The authors declare that they have no competing interests.

Olvija Komasilova conceived and designed the experiments, performed the experiments, analyzed the data, prepared figures and/or tables, authored or reviewed drafts of the paper, and approved the final draft.

Vitalijs Komasilovs conceived and designed the experiments, analyzed the data, authored or reviewed drafts of the paper, and approved the final draft.

Armands Kviesis analyzed the data, authored or reviewed drafts of the paper, and approved the final draft.

Aleksejs Zacepins conceived and designed the experiments, performed the experiments, analyzed the data, prepared figures and/or tables, authored or reviewed drafts of the paper, and approved the final draft.

The following information was supplied regarding data availability:

The code is available in the Supplementary Files.

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
