# Peer review of "Model for finding the number of honey bee colonies needed for the optimal foraging process in a specific geographical location"

_PeerJ, doi:10.7717/peerj.12178_

## Round 0.1 · original submission · Major Revisions

Please address the reviewer's comments.

In particular, the results and discussion sections need some work. The flow of the model description is hard to follow, thus making it difficult to understand or replicate and the section presented as results and discussion should be moved to the section materials and methods. A re-framed discussion section should place your results in the context of the previous literature, current relevance and discuss the limitations of your work.

I look forward to receiving the revised manuscript.

Reviewer 1 ·

Basic reporting

The authors used a clear and unambiguous professional English throughout the manuscript. I suggest adding some citations and literature references to support some statements and fulfill background/context information.
This manuscript sheds light on solving an important problem for beekeeping. I understand that this is not a definitive and precise model, but it has its finding and in the future can help to improve the productive levels in beekeeping. I missed a well-elaborated discussion comparing it with other relevant works dealing with the same theme, as well as comparing it with other techniques for estimating the ideal number of colonies per apiary.

Experimental design

The research question is well defined, relevant and significant. However, for the future, it would be interesting to validate it in real life, choosing some apiaries to estimate the ideal bee colonies. In general, methods are described in sufficient detail and information to replicate. Furthermore, math models leave possibilities for adding more variables that may be necessary to refine the calculation in another region or depending on the context.

Validity of the findings

The conclusions are well presented, linked to the original research question, and limited to supporting results. However, I would suggest the authors better define what does this mean: “the number of bee colonies needed for the process of ideal foraging”. Does it refer to honey production or colony maintenance? Some claims sound like ordinary beekeeping procedures or even speculation rather than a scientific method. Please add solid and recent references to support these statements.

Additional comments

- Scientific name must be in italics - Apis mellifera
- Please add references to support some statements in the Introduction Section e.g. " ...location usually is based on beekeepers’ experience or the limited availability of physical space." (lines 48-49), "..In most cases, success in beekeeping depends on the availability of sufficient bee foraging resources in
terms of both quality and quantity of nectar and pollen." (lines 64-65)
- In the Introduction, the authors use examples for honeybee colonies densities that occur naturally in forests. This number can vary widely from biome to biome and from agroecosystem to agroecosystem. So, apiaries mean colonies managed by beekeepers. Thus, I would suggest the authors use examples of carrying capacity of areas for colonies managed in apiaries to be closer to the objective of this work.
- Line 50-51: This statement is confusing. There is a vast literature that defines the number of bee colonies per area that must be introduced into a field/orchard or plantation to maximize production through pollination, according to the biology of the crop species. Please see Delaplane et al. (2013)- Standard methods for pollination research with Apis mellifera. So, please, consider rewriting this statement.
- Please provide a citation and some evidence to support this number of 0.35, further explaining what this bee foraging efficiency index would be. Would it be the ratio of energy gained to energy expended to acquire food? Is this related to the Optimal Foraging theory? Please, provide more details.
- Fig 9d and Fig 9e: What is the relevance of these variable/map for the formula? Please, explain better. Provide more details.
- In Section 3.9. Calculation of a number of colonies to be placed at a specific location (x,y), the authors should explain the calculation step by step and add discussion to it. Compare with other previous publications and different techniques that estimate the optimal number of colonies for an apiary. Improve this section!
- The numbers in the figure captions are different from the figure numbers in the body of the text. So check everything again, choose the most suitable one and align the number in the Captions to the body of the text.

·

Basic reporting

1. While the manuscript is written in an understandable manner, there are some issues with the English. The use of articles (e.g. in the title "in the specific geographical location" should be "in a specific geographical location"), in particular, should be reviewed. There are other instances where phrasing could be better (e.g. in line 71 - "the previous author model" could instead be written "the authors' previous model"; in line 88 - "road effect on the foraging activity" could be written "effect of roads on foraging activity").

2. Some of the notation used in the equations is, to my experience, non-standard. In particular, equations (8) and (9), which I believe to be denoting functions transforming fields and roads into polygons and lines respectively, should be written using more widely used standards of notation (e.g. https://en.wikipedia.org/wiki/Function_(mathematics)#Notation ).

Moreover, the notation for discrete uniform distributions "R" (first introduced in line 232) is non-standard (see https://en.wikipedia.org/wiki/Discrete_uniform_distribution ). Replacing "R" with the standard "U" notation would be especially helpful given that R is also used in line 265 to represent "roads from regions of interest".

3. Explanation of all variables in equations should be provided. For example, what is P_n in equation (8)?

4. Typo in line 261: "," in "-0,1" should be a ".".

5. Though it builds on previous work by the authors, the work is self contained. Sufficient background literature is provided. The figures are clear.

Experimental design

1. The formulation of "optimal foraging performance" seems slightly off. In like 45-47, the authors describe the number of hives that gives "optimal foraging performance" as the "number of colonies placed at one geographical location (within one apiary)" such that "[the colonies] should be able to forage the maximal amount of potential resources available". I believe this definition to be incomplete as, under said definition, the number of hives can increased indefinitely until all the potential resources are used up (that is, the colonies forage the "maximal amount of potential resources available" once they forage all available resources). I reckon the definition should be revised to be more in line with the definition of "optimal location" seen in the authors' previous work (Komasilova et al., 2020): "optimal location will allow bee colonies to forage on higher amount of the resources with minimal energy consumption." (the key bit being the minimisation of energy consumption).

2. The authors tackle a problem that is meaningful to beekeepers. The method proposed seems sensible. Code is provided and the description of the method is detailed.

Validity of the findings

The model is evaluated on dummy data and has not been evaluated under real-world conditions. It is thus difficult to say how well the model proposed by the authors would function in practice. This limitation is mentioned by the authors.

Reviewer 3 ·

Basic reporting

The presented manuscript addresses a relevant question in modern beekeping. The general readivility can be improved if more attention is paid to lenguage details. Some phares can be improved by adding puntuation marks.

The presented background is generally good, althouhg it can be improved making more enphasis on why the research is needed. For example, some lines can be added to explain what is the current problem when honeybee hives are introduced without a proper knowledge of the carrying capacity of the landscape.

The section matherials and methods can use a major transformation to facilitate its readibility as the current version is difficult to follow. Equations are not referenced in the main text and some terms are not defined. Hence it is not clear the meanining of the equations. I suggest the authors to consult the existing litterature presenting related models (for example the beehave model) to get an idea on how a model can be propertly described. In general, further work is needed to improve the flow of this section.

The section results and discussion are not property labeled. In this section the model was further described and presented with an example. This information should be in the section of materials and methods rather than in results and discussion.

Experimental design

The research question is well defined, relevant and meaninigful. With some exta work, it can clearly state how this research fills the identified knowledge gaps. However, the content of the manuscript does not seems to represent a thorough research. It does not have a propertly stablished result section and it does not have a discussion sections that frames the current work in the litterature (existing models, implication of finidings, concrete further research).

Validity of the findings

The impact of the research in the current scientific context is not assessed. The presented work is not replicable under the current model description. It is not explained how the values used in the model were obtained. In order to make the model replicable, further work needs to be done for improve the flow of the description.

The research lack results and discussion to elaborate further the conclusions.

---

## Round 0.2 · accepted · Accept

Hi there, I have reviewed the manuscript and rebuttal letters and feel it can now be accepted for publication. Thank you for your revisions.